# A Simulation Study on the Interaction Between Pollutant Nanoparticles and the Pulmonary Surfactant Monolayer

**DOI:** 10.3390/ijms20133281

**Published:** 2019-07-04

**Authors:** Kai Yue, Xiaochen Sun, Jue Tang, Yiang Wei, Xinxin Zhang

**Affiliations:** School of Energy and Environmental Engineering, University of Science and Technology Beijing, Beijing 100083, China

**Keywords:** air-pollutant nanoparticle, coarse-grained model, interaction, molecular dynamics simulation, pulmonary surfactant monolayer

## Abstract

A good understanding of the mechanism of interaction between inhaled pollutant nanoparticles (NPs) and the pulmonary surfactant monolayer is useful to study the impact of fine particulate matter on human health. In this work, we established coarse-grained models of four representative NPs with different hydrophilicity properties in the air (i.e., CaSO_4_, C, SiO_2_, and C_6_H_14_O_2_ NPs) and the pulmonary surfactant monolayer. Molecular dynamic simulations of the interaction during exhalation and inhalation breathing states were performed. The effects of NP hydrophilicity levels, NP structural properties, and cholesterol content in the monolayer on the behaviors of NP embedment or the transmembrane were analyzed by calculating the changes in potential energy, NP displacement, monolayer orderliness, and surface tension. Results showed that NPs can inhibit the ability of the monolayer to adjust surface tension. For all breathing states, the hydrophobic C NP cannot translocate across the monolayer and had the greatest influence on the structural properties of the monolayer, whereas the strongly hydrophilic SiO_2_ and C_6_H_14_O_2_ NPs can cross the monolayer with little impact. The semi-hydrophilic CaSO_4_ NP can penetrate the monolayer only during the inhalation breathing state. The hydrophilic flaky NP shows the best penetration ability, followed by the rod-shaped NP and spherical NP in turn. An increase in cholesterol content of the monolayer led to improved orderliness and decreased fluidity of the membrane system due to enhanced intermolecular forces. Consequently, difficulty in crossing the monolayer increased for the NPs.

## 1. Introduction

Studies show that high concentrations of fine particulate matter (PM2.5) is the underlying cause of haze. The surfaces of PM2.5 particles can carry many harmful organic and inorganic molecules, and the particles can readily enter the body via inhalation and ingestion primarily through the respiratory tract. The inhaled particles may deposit near the alveoli and interact with the pulmonary surfactant monolayer, which is used to regulate the surface tension of the lungs and, thus, maintain the stability of alveoli [1,2]. This interaction and particle invasion may affect the normal function of alveoli [3,4] and even cause acute lung injury [5], cardiovascular disease, inflammation of the respiratory tract, and hematological toxicity [6]. Meanwhile, nanoparticles (NPs) have significant advantages over normal bulk materials owing to their unique optical, photothermal, and electromagnetic properties. NPs can easily cross biological barriers and access specific intracellular locations, thereby enabling many potential biomedical applications such as sensing and imaging, drug delivery, and hyperthermia. Accordingly, NPs with membrane-penetration abilities and high delivery efficiencies should be designed. 

The interaction between the pulmonary surfactant monolayer and inhalable particles is the most primitive bio-nano process. Understanding this interaction at the molecular level is one of the most important steps in studying the impact of NPs on the human body and can also provide guidance for NP-based clinical applications. Some experiments and simulations have been carried out, and many advances have been made [7,8,9,10]. The physicochemical properties of NPs reportedly play an important role in influencing their interaction with the pulmonary surfactant monolayer. Experimental results show that NP hydrophobicity increases their retention in the monolayer [11], and large NPs have a greater degree of damage to the monolayer [12]. Cholesterol can enhance the interaction between the monolayer and drug and, thus, inhibit drug penetration [13] when the cholesterol content in the monolayer reaches 20%. Some simulations have been performed using molecular dynamics methods, which are widely used to compute biomolecular interactions on micro/nano scales by determining the positions and velocities of particles in terms of the laws of classical physics [14]. The results of analysis of the equilibrium position of NPs, potential of mean force, and order parameters indicate that the structural or morphological properties of the monolayer are obviously affected by the structural and surface physicochemical properties of NPs [15,16,17,18,19]. However, most simulation studies have been carried out based on the generalized coarse-grained (CG) NP models and the CG models of a pure dipalmitoylphosphatidylcholine (DPPC) monolayer or cholesterol-free monolayer. The pulmonary surfactant monolayer comprises phospholipids and proteins, wherein approximately 90% of the monolayer is phospholipids and 10% is proteins. The main component of phospholipids is DPPC with a small amount of neutral lipids (5%~10%), and cholesterol constitutes the major fraction of neutral lipids [20]. Proteins include four surfactant proteins (i.e., SP-A, SP-B, SP-C, and SP-D). Among them, SP-A and SP-D are macromolecular proteins dominating the metabolism and immunoregulation of the monolayer [21], whereas SP-B and SP-C are small molecular proteins dominating the stability of their surface-sustaining monolayer [22].

In the present study, we used molecular dynamics simulations to investigate the interaction mechanism of NPs with the pulmonary surfactant monolayer and the behaviors of NP translocation across the monolayer at exhalation and inhalation breathing states for the representative atmospheric particulate matters. The coarse-grained (CG) model of DPPC containing cholesterol molecules and the surfactant-specific proteins SP-B and SP-C were established, and a qualitative comparison of the simulation results and experimental data from literature was carried out. On this basis, we comprehensively analyzed the influence of crucial factors affecting NPs and the monolayer such as NP type, shape, size, hydrophilicity/hydrophobicity, and cholesterol content on NP interactions and translocation. 

## 2. Results and Discussion

To investigate the interaction mechanism of four kinds of representative NPs with the pulmonary surfactant monolayer during natural breathing patterns, we analyzed the mean-square displacement (MSD) of the NPs, centroid-to-centroid distance between the NP and monolayer, order parameter of the phospholipid molecule, and surface area per lipid molecule in the interaction. The MSD is used to evaluate the movement of a given particle, indicating its diffusion rate [23]. It is calculated with the following formula:(1)$$MSD=R\left(t\right)=\{|\vec{r}\left(t\right)-\vec{r}\left(0\right){|}^{2}\},$$
where $\vec{r}\left(t\right)$ is used to represent the position of a particle at time *t*.

The order parameter P_2_ is defined as [24]
(2)$$P_{2}=\frac{1}{2}{(3\cos}^{2}\theta -1),$$
where θ represents the angle between the bond linking two CG beads and the *z*-axis, and these two beads are DPPC C3B and C4B.

### 2.1. Interactions of Different Nanoparticles (NPs) with the Pulmonary Surfactant Monolayer in Exhalation and Inhalation Breathing States

Figure 1a,b show the snapshots of interactions among the different NPs and the pulmonary surfactant monolayer during compression and expansion, respectively. Water molecules are not shown in the figures to emphasize NP movement. We can observe that C and CaSO_4_ NPs were embedded into the monolayer at the exhalation breathing state, whereas C_6_H_14_O_2_ and SiO_2_ NPs penetrated the monolayer. Compared with CaSO_4_, C was almost completely wrapped by the monolayer and caused the monolayer to bulge, prominently, toward to the water phase, generating a large monolayer curvature. Thus, the interaction of C can inflict great disturbance to the monolayer structure. Figure 1b indicates that C_6_H_14_O_2_ and SiO_2_ NPs still translocated across the pulmonary surfactant monolayer throughout the inhalation breathing process as they did throughout the exhalation breathing process. However, unlike in the exhalation breathing state, CaSO_4_ penetrated the monolayer, although C NP still could not cross and was embedded into the monolayer.

The MSDs of the NPs and the centroid-to-centroid distances between the NPs and the monolayer during compression and expansion were calculated, respectively, as shown in Figure 1c–f. In the exhalation breathing state, the MSD of SiO_2_ was larger than those of the other three NPs, and the MSD of C was higher than that of CaSO_4_ NP, indicating that the diffusion rate of the SiO_2_ NP was the fastest, and C moved a longer distance than CaSO_4_. Figure 1e also shows that C and CaSO_4_ NPs reached a relatively equilibrated state within approximately 10 ns, whereas C_6_H_14_O_2_ and SiO_2_ NPs took a longer time (i.e., around 45 ns). The initial centroid-to-centroid distance between the NP and the monolayer at time *t* = 0 was set to 3.2 nm. The distance of the centroid position between the CaSO_4_ NP and the monolayer at the equilibrium state was about 1.52 nm, and C was 0.27 nm away from the monolayer, meaning that C was embedded in the monolayer more deeply than CaSO_4_. The final positions of C_6_H_14_O_2_ and SiO_2_ NPs were all approximately −3.3 nm away from the monolayer, with an average thickness of 2.08 nm, indicating that both NPs completely translocated across the monolayer. Compared with the compression process, the MSDs of all NPs during expansion were larger, which meant that all NPs could move faster, and their diffusion abilities were stronger than those in the exhalation breathing state. Another difference was that the MSD of CaSO_4_ was larger than that of C, although SiO_2_ showed the largest MSD. Figure 1f shows that C_6_H_14_O_2_ and SiO_2_ NPs reached equilibrium within approximately 40 ns, which was shorter than that in the exhalation breathing state. Notably, the final position of the C NP was about 0.88 nm away from the centroid of the monolayer in the inhalation breathing state, indicating the C NP was embedded into the monolayer more deeply in the exhalation breathing state. These analysis results of MSD and the centroid-to-centroid distance confirmed our previous findings obtained from the molecular dynamics simulation snapshots.

The differences in hydrophilicity/hydrophobicity of the NPs explained the difference among the interaction behaviors. C NPs are typical hydrophobic NPs, and CaSO_4_ NP is semi-hydrophilic. C_6_H_14_O_2_ NPs and SiO_2_ NPs are strongly hydrophilic NPs, and SiO_2_ NPs are more hydrophilic.

The strong hydrophilicity of C_6_H_14_O_2_ and SiO_2_ NPs led to the strong attractive forces between the NPs and the hydrophilic head groups of DPPC molecules or between the NPs and the water molecules on the other side of the monolayer, which enabled the NPs to overcome the energy barrier and translocate across the monolayer. The uncharged and hydrophobic C NP was attracted by the hydrophobic tail groups of DPPC molecules, and its interaction with the hydrophilic head group was repulsive at the same time. Consequently, C cannot translocate across the monolayer, and its deep encapsulation inside the monolayer caused monolayer deformation.

For the semi-hydrophilic CaSO_4_ NP in the exhalation breathing state, the hydrophobic interaction between the NPs and hydrophobic apolar tail groups of DPPC molecules were not intensive. The interactions between the NP and hydrophilic head groups and water molecules were attractive but not sufficiently large for NP penetration, only causing the NP to stay in the surface region of the monolayer. In the inhalation breathing state, the reason for CaSO_4_ translocation across the monolayer may be that the relatively loose arrangement of lipid molecules enabled the attractive interaction between the NP and the monolayer to overcome the energy barrier, resulting in NP penetration through the expanded monolayer. We further elucidated this aspect by analyzing the surface area per lipid molecule and the energy component of the simulation system, as discussed in the following sections.

We then calculated the order parameters of lipid molecules in the pure monolayer system and NP monolayer systems over time, as well as the surface area per lipid molecule, to analyze the effect of interaction of NPs with the monolayer on the structural properties of the monolayer. Figure 2a,b shows the changes in order parameters and surface area per lipid molecule in the exhalation and inhalation breathing states, respectively. A larger order parameter represented better consistency between the bonding and arrangement directions of the phospholipid monolayer, indicating better orderliness of the membrane lipid molecules. Figure 2a shows that the order parameters of lipid molecules in the systems containing C_6_H_14_O_2_ and SiO_2_ were almost the same as those in the system without NPs, either in exhalation or at inhalation breathing states, all of which were approximately 0.458 or 0.285 (Figure 2a,b) at equilibrium, respectively. Thus, C_6_H_14_O_2_ and SiO_2_ penetrations had a negligible effect on the orderliness of lipid molecules after NP membrane translocation. The order parameters for the system containing CaSO_4_ NPs were 0.452 and 0.285 in exhalation or at inhalation breathing states, respectively, but they were 0.413 and 0.204 for C, indicating that the penetration of CaSO_4_ had little effect on the monolayer structure in the inhalation breathing state. The presence of C most obviously affected the monolayer structure and led to decreased orderliness in the monolayer.

Figure 2c,d shows that the surface areas per lipid molecule in the systems without NPs and with C_6_H_14_O_2_ and SiO_2_ NPs were all about 0.53 nm^2^ after equilibrium was reached. In other words, the effect of C_6_H_14_O_2_ and SiO_2_ on the monolayer structure was almost negligible. The surface areas per lipid molecule of the system containing C was about 0.552 and 0.657 nm^2^ in exhalation or inhalation breathing states, respectively. These values were noticeably larger than those of other systems because the hydrophobicity of C NPs produced remarkable monolayer curvature and a large change in area. Analysis results of surface areas per lipid molecule were consistent with those of the order parameters. In general, the difference in interactions with the monolayer between SiO_2_ and C_6_H_14_O_2_ was not obvious because the differences in their hydrophilicity levels was not large.

### 2.2. Analyses of System Energy and Surface Pressure of the Monolayer

We calculated the total energy of different systems to compare the interactions of different NPs with the pulmonary surfactant monolayer in inhalation and exhalation breathing states and to further understand the mechanism underlying such interactions. All simulations in this study were performed in the NPT ensemble, and the total kinetic energy of the system remained unchanged considering the constant temperature under practical conditions. Thus, the change in total energy was equal to the change in potential energy. 

As shown in Figure 3a, for the systems with the same NP, the total energy of the system in the inhalation breathing state was lower than that in the exhalation breathing state (i.e., the orderliness of the lipid molecules was more orderly, and the entropy of the system was lower). For all NPs, the total energies in both breathing states decreased after interacting with the monolayer, indicating that the interaction was irreversible. During the interaction process, the potential energy was converted to kinetic energy and was then dissipated by the simulation system to keep the total kinetic energy constant, so the potential energy of the system decreased. The NPs could not enter the air side again without applying external disturbance after it was embedded into the monolayer or crossed the membrane. Figure 3b shows the difference in total energy between the NP monolayer system and the pure monolayer system, which includes the energy of the NP, the interaction energy between the NP and monolayer, and the interaction energy between the NP and water molecules. The change in this difference can reflect the change in NP potential energy, owing to the constant kinetic energy in the NPT ensemble. We also found SiO_2_ had higher potential than C, indicating a larger interaction of SiO_2_ with the lipid and water molecules and a greater likelihood of NP crossing the monolayer. The potential energy of SiO_2_ in the inhalation breathing state was larger than that in the exhalation breathing state, indicating that the NP was subject to greater force and could more easily penetrate the monolayer.

The attractive interaction between the hydrophobic NP and hydrophobic groups of the monolayer was the dominant driving force acting on the NP during the process of NP movement and embedment, which had the same direction as the NP motion. Accordingly, the potential energy continually decreased until it eventually stabilized at a position where it was wrapped and a potential well formed. Conversely, hydrophilic NPs were subjected to a repulsive force, owing to the existence of a hydrophobic barrier produced by the lipid monolayer. The NP overcame the potential barrier and finally penetrated the monolayer when the hydrophilic interaction between the NP and water molecules was sufficiently strong. Otherwise, the NP was stopped by the potential barrier and embedded into the monolayer.

A stable, low surface tension of the pulmonary surfactant monolayer can maintain the stability of lung alveolars [1,20], but the surface tension may be affected by the interaction between the NPs and the monolayer. We calculated the surface pressure and pressure–area isotherm of the monolayer, which was commonly used to analyze the functional properties of the monolayer. The surface pressure π is given by π = *γ_w_* − *γ*, where *γ_w_* and *γ* represent the surface tension of pure water and the monolayer, respectively.

Figure 3c shows the changes in surface pressure of the monolayer in the exhalation and inhalation breathing states for the pure monolayer and NP monolayer systems after reaching the equilibrium state. Compared with the surface tension of the monolayer in the system without NPs, all interactions of different NPs with the monolayer resulted in decreased surface tension of the monolayer to a certain extent. The effects of all interactions in the exhalation breathing state were more remarkable than those in the inhalation breathing state, in which C showed the most pronounced effect on the surface tension of the monolayer. C was embedded into the monolayer because of its hydrophobic interaction with the hydrophobic tail chain of DPPC molecules and its interaction with the hydrophilic head groups of the lipid. This phenomenon resulted in decreased free area available for the lipid molecules and increased surface density of the monolayer. Thus, the surface tension of the monolayer was reduced and the surface pressure increased. For CaSO_4_, although it was also embedded in the monolayer in the exhalation breathing state, the effect of the interaction on the surface pressure of the monolayer was still smaller than that of C because it had a smaller embedding degree, as shown in Figure 1a. In the inhalation breathing state, the effect of the interaction of CaSO_4_ with the monolayer was very slight, similar to those of C_6_H_14_O_2_ and SiO_2_ NPs, to their complete membrane translocation. These findings indicated that the breathing state played an important role in the interaction of NPs with the monolayer.

We compared the calculated surface pressure–area isothermal (π-A) of the monolayer for the strongly hydrophilic SiO_2_ NP and the hydrophobic C NP with experimental data obtained from Ref. [25] (Figure 3d). Qualitative agreement was observed between the simulation and experimental results (i.e., strongly hydrophilic NPs had a smaller effect on the surface tension of the monolayer than the hydrophobic NPs). This finding was due to ability of the strongly hydrophilic NPs to translocate across the monolayer and not stay in it at equilibrium. The reason for the quantitative difference in simulation and experimental results was that the NPs initially mixed with DPPC for compression in the experiment, whereas the NPs were initially located on the air-side in the simulation and took more time to move to and interact with the monolayer.

### 2.3. Effect of Cholestrol Content of the Monolayer and NP Structural Properties on the Interaction

Cholesterol is reportedly present in the lipid membrane, at 5%–10% by mass, and constitutes the major fraction of neutral lipids [3]. To investigate the effect of cholesterol content on the interaction of NPs with the pulmonary surfactant monolayer, we established CG models of the pulmonary membrane with 0%, 5%, and 10% cholesterol content, respectively, and simulated its interaction with C, CaSO_4_, or SiO_2_ NPs. We found that all kinds of NPs moved slower with increased cholesterol content, and that C could not penetrate all three kinds of the monolayers at the inhalation breathing state, whereas SiO_2_ could cross all these monolayers. 

However, the calculation results of the centroid–centroid distance of the NP with the monolayer and the MSD of the lipid molecules (Figure 4a,b) revealed that CaSO_4_ can translocate only across the monolayer with 0% and 5% cholesterol content, and the MSD of the lipid molecules prominently decreased in the system with 10% cholesterol content. We then analyzed the order parameters of the lipid molecules of the monolayer with 0%, 5%, and 10% cholesterol contents, respectively, in the systems without NPs and with CaSO_4_ (Figure 4c,d). An increase in cholesterol content of the monolayer led to an increase in the order parameters of the monolayer. The fluidity of lipid molecules is usually related to their structural ordering and is manifested in the tail of phospholipid molecules [26]. Our results indicated that the cholesterol content remarkably influenced the fluidity of the pulmonary surfactant monolayer and that increased cholesterol content led to a more orderly arrangement of lipid molecules. This ordering enhanced the packing of the hydrophobic portion of the lipid molecule and, thus, reduced the fluidity of the pulmonary surfactant monolayer. Subsequently, the NPs encountered increased difficulty in penetrating the monolayer. CaSO_4_ could cross the monolayers with 0% and 5% cholesterol content, so little difference in order parameter between the pure monolayer and CaSO_4_ NP monolayer systems was observed. For the lipid molecules of the monolayer with 10% cholesterol content, the order parameters of the two systems were 0.40 and 0.42, respectively. This finding indicated that the difference was not very remarkable because CaSO_4_ stayed only on the surface region of the monolayer, and it inflicted slight disturbance to the monolayer structure because the centroid–centroid distance between the NP with 3 nm diameter and the monolayer with 2 nm thickness was only 2.23 nm.

We analyzed the effects of NP structural properties on the interaction between NPs and the pulmonary surfactant monolayer, taking the effects of size and shape into account. Figure 5a shows the centroid-to-centroid distances between a hydrophilic and spherical NP without charge and surfactant monolayer and the corresponding snapshots at equilibrium under different NP size conditions, respectively. Smaller NPs took shorter times to cross the monolayer and reach the equilibrium state, indicating that the difficulty of NP membrane translocation was positively correlated with NP size. Notably, all NPs attached onto the hydrophilic side of the monolayer after membrane translocation, and no detachment from the monolayer occurred due to the attractive interaction of the hydrophilic head groups of the monolayer with the NPs. Although all hydrophilic NPs can successfully cross the monolayer, the NPs did not stay in a certain position and remain fixed, and the smaller NPs were more easily affected by thermal fluctuations and had a larger fluctuation range. 

Three types of neutral hydrophilic NPs with three common shapes for NPs in the air, namely, spherical (diameter d = 3 nm), rod-shaped (diameter d = 3 nm and height h = 3 nm), and flaky (size d = 3 nm and thickness δ = 0.5 nm), were modeled to analyze the effect of NP shape on the interaction. As shown in Figure 5b, all three types of NPs can translocate across the monolayer. The difference was that the spherical and rod-shaped NPs still adsorbed onto the monolayer at equilibrium, whereas the flaky NP penetrated through and completely separated from the monolayer, entering into the water phase. During NP translocation, we observed that the angles between the monolayer and rod-shaped or flaky NPs were continuously adjusted according to the changes in their relative position and the variation in their interactions. The rod-shaped NP contacted and penetrated the monolayer with the long axis perpendicular to the plane of the monolayer from the initial parallel placement. Similarly, the flaky NP spontaneously moved and rotated from the initial placement in parallel with the monolayer to the vertical placement with a minimum contact area made with the monolayer. After the NP was embedded into the monolayer, the NP eventually detached from the monolayer because attachment of the NP to hydrophilic head groups of the monolayer cannot counteract the attractive force between the NP and water molecules due to minimal contact. 

To quantify the difference in penetration behaviors of NPs with different shapes, we calculated the centroid-to-centroid distance between NPs and the monolayer, and we found that rod-shaped NPs took less time (t = 17 ns) to cross the monolayer and reach the equilibrium state than spherical NPs (t = 21 ns), although all of them eventually attached onto the monolayer and could not detach from it. The flaky NP spent more time (t = 50 ns) to penetrate the pulmonary surfactant monolayer, and equilibrium was reached within about 90 ns, where it then completely entered the water phase and was surrounded by water molecules.

## 3. Methods

All simulations were carried out using the MARTINI [27,28] CG force field, in which several heavy atoms were incorporated into one bead to decrease computational cost as well as allow larger length scale and longer time-period simulations compared with all-atom models. The coarse-graining procedure would reduce the resolution of the system and might cause some key information at the atomic scale to be lost [29]. In our simulation, the model of pulmonary monolayer was established using the standard coarse-grained MARTINI force field, and the coarse-graining of NPs was performed by using one bead to represent one atom for reducing the loss of information. 

The CG model of the DPPC pulmonary surfactant monolayer containing cholesterol and surfactant-specific proteins (i.e., SP-B and SP-C) was established using GROMACS [30] and Visual Molecular Dynamics [31], as shown in Figure 6a–d. Given that organic carbon, elemental carbon, SO_4_^2−^, and NO_3_^−^ are the most abundant chemical components of PM2.5 [32], we selected C, C_6_H_14_O_2_,and CaSO_4_ NPs to represent elemental carbon, alcoholic content, and secondary sulfate particulates discharged from automobile exhaust, coal combustion, and biomass burning, and we selected SiO_2_ to represent the main inorganic component of wind-blown soils. The CG models of these four kinds of NPs were constructed using GROMACS and MATERIAL STUDIO (Figure 6e). Relevant information about bond length and bond angle of the NPs was obtained by implementing the MSI2LMP tool packaged with LAMMPS software. To ensure rigidity of the NPs during simulation, the equilibrium distance among beads was set to 0.47 nm, and the force constant was set to 1250 kJ/mol. 

The model of the pulmonary surfactant monolayer consisted of 1086 CG DPPC molecules, 34 CG cholesterol molecules, and 2 CG surfactant proteins (SP-B and SP-C). They were placed in a 20 × 40 × 80 nm cubic simulation box containing 46634 CG water molecules and one NP. The monolayer was placed on the liquid–air interface in the z-direction, and the NP was initially located in the air above the monolayer. The general shape of the NP was spherical, and its diameter was approximately 3 nm. The simulation box was stretched along the z-direction to prevent water molecules in the system from running through the periodic boundary to the airside of the monolayer, as shown in Figure 6f.

In the simulation, a cutoff distance of 1.2 nm was set for van der Waals interactions, and the Lennard–Jones potential was smoothly shifted to zero in the range of 0.9–1.2 nm to reduce cutoff noise. For electrostatic interactions, the Coulomb potential had a cutoff distance of 1.0 nm and a smooth shift to zero over the range of 0–1.0 nm. An isobaric–isothermal (NPT) ensemble with periodic boundary conditions was used for simulations. The temperature of the system was controlled at 310 K using the Berendsen thermostatting method, and semi-anisotropic coupling was used by the Berendsen barostat with a coupling constant of 3.0 ps and a compressibility of 4.5 × 10^−5^ bar^−1^ in the *x*–*y* plane and 0 bar^−1^ on the *z*-axis. Given that the respiratory scale is several orders of magnitude larger than the nanoscale of the molecular simulation study [33,34], a large lateral pressure was applied in the simulation to achieve the surface tension of the pulmonary surfactant monolayer in the respiration process, which was 30% higher and lower than that at initial state for the exhalation (compression) and inhalation (expansion) breathing states, respectively. We implemented postprocessing and visualization through VMD, and all simulations were performed using the Gromacs 4.5.4 simulation package. The simulation time of the system was set to 150 ns.

## 4. Conclusions

This work investigated the interaction mechanism between the representative pollutant NPs and the pulmonary surfactant monolayer using molecular dynamics simulations based on established CG models. By analyzing the snapshot, NP MSD and centroid distance, the order parameter and area of the lipid molecules, system energy, and surface pressure of the monolayer, we concluded that the strongly hydrophilic SiO_2_ and C_6_H_14_O_2_ NPs can cross the monolayer in either exhalation or inhalation breathing states and had little effect on the monolayer structure at equilibrium, whereas NP penetration of the semi-hydrophilic CaSO_4_ NP occurred only in the inhalation breathing state with a slight influence on monolayer orderliness. No obvious difference was observed in NP membrane translocation and effect on the monolayer between SiO_2_ and C_6_H_14_O_2_ NPs because there was a slight difference in hydrophilicity levels. 

The hydrophobic C NP had the most pronounced effect on monolayer orderliness because it could not cross and embedded only into the monolayer during both breathing states. The difficulty in NP membrane translocation for the hydrophilic NPs was positively correlated with the cholesterol content in the monolayer because of the change in fluidity caused by cholesterol (e.g., the semi-hydrophilic CaSO_4_ NP could not penetrate the monolayer with 10% cholesterol content in the inhalation breathing state). The flaky shape had the advantage of membrane translocation for the hydrophilic NPs over the spherical and rod-shape ones, and only the hydrophilic flaky NP could detach from the monolayer and enter the water phase completely. The interaction of the hydrophilic SiO_2_ NP had a relatively slight effect on the surface tension of the monolayer, whereas that of the hydrophobic C NP could obviously decrease the surface tension. These findings indicated that NP deposition could affect its ability to regulate interfacial tension. Our simulations can provide information on the effect of NP hydrophilicity, cholesterol content of the monolayer, and NP structural properties on the interactions of air-pollutant NPs with the pulmonary surfactant monolayer, and the knowledge can be useful for drug-delivery design.

## Figures and Tables

**Figure 1 ijms-20-03281-f001:**
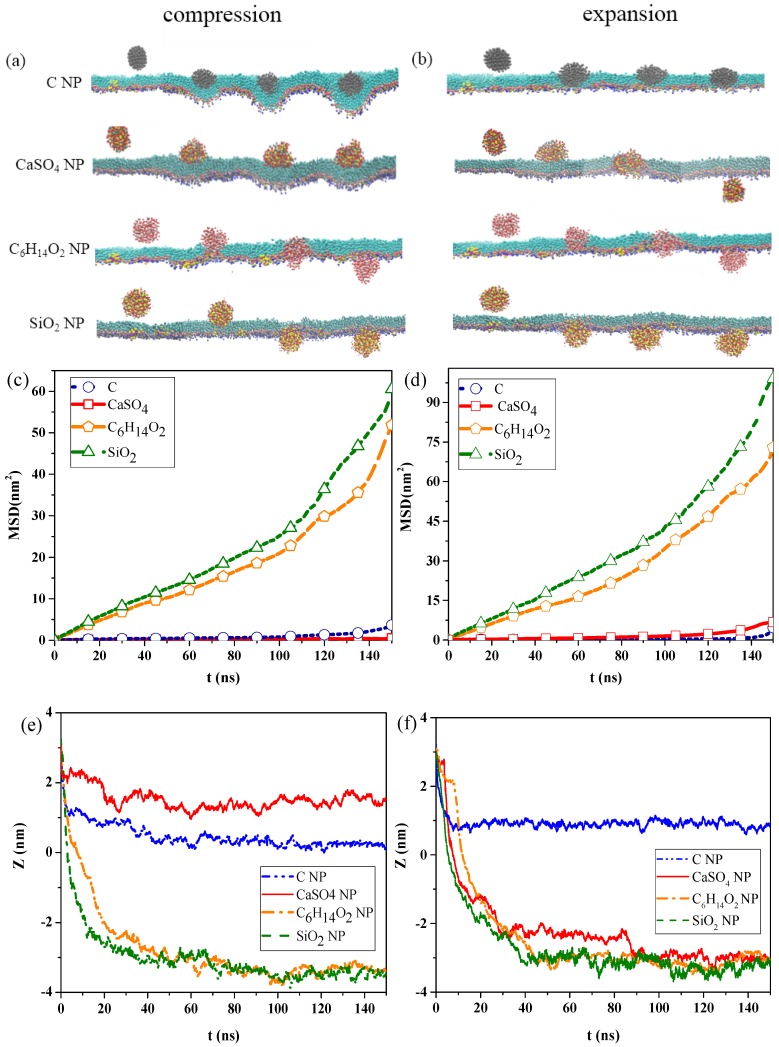
Interaction between nanoparticles (NPs) and the pulmonary surfactant monolayer. Snapshots of the interaction in the (**a**) exhalation and (**b**) inhalation breathing states; mean-square displacements (MSDs) of the NPs in (**c**) exhalation and (**d**) inhalation breathing states; centroid-to-centroid distance between the NPs and monolayer in (**e**) exhalation and (**f**) inhalation breathing states.

**Figure 2 ijms-20-03281-f002:**
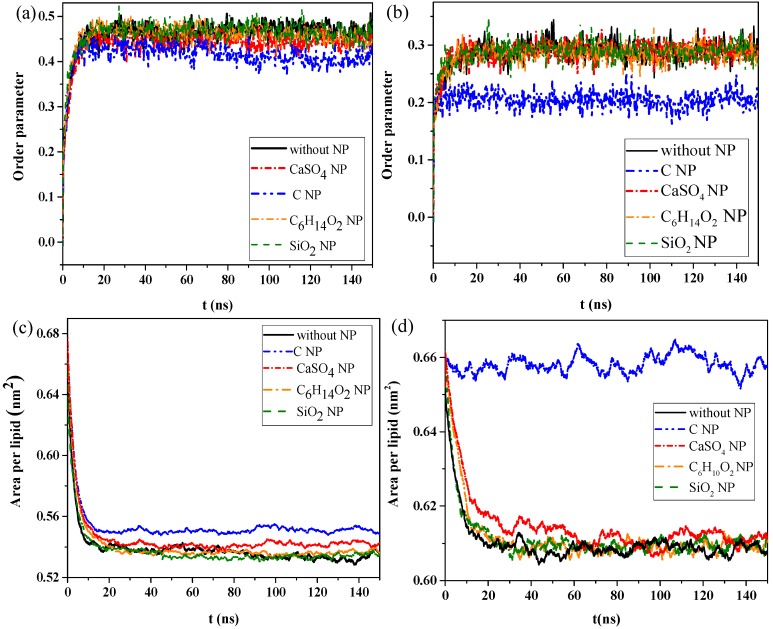
Order parameters of monolayer molecules in the (**a**) exhalation and (**b**) inhalation breathing states. Surface area per lipid molecule in the (**c**) exhalation and (**d**) inhalation breathing states.

**Figure 3 ijms-20-03281-f003:**
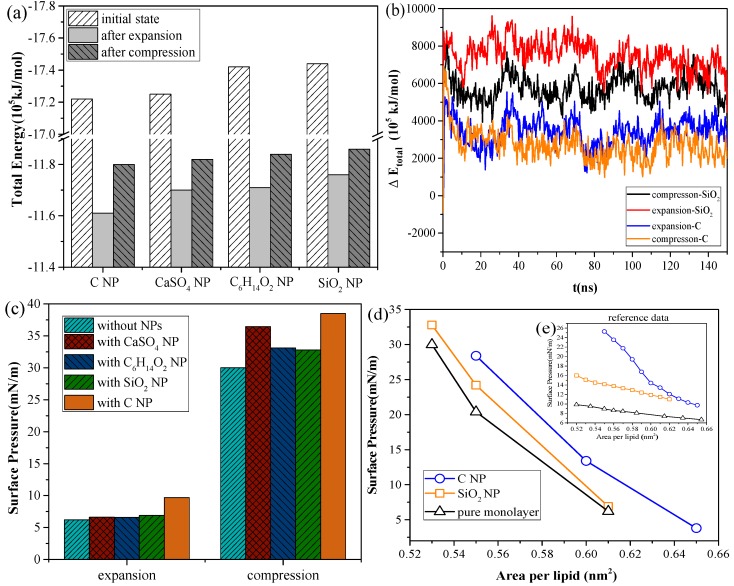
Analyses of system energy and surface pressure for the systems containing a pure monolayer or an NP monolayer: (**a**) total energy of different systems after interaction, (**b**) ΔE_total_ (i.e., the difference in total energy between the NP monolayer system and the pure monolayer system), (**c**) surface pressure of the monolayer in exhalation and inhalation breathing states, and (**d**) surface pressure–area isotherms of the monolayers from the simulation and Ref. [25].

**Figure 4 ijms-20-03281-f004:**
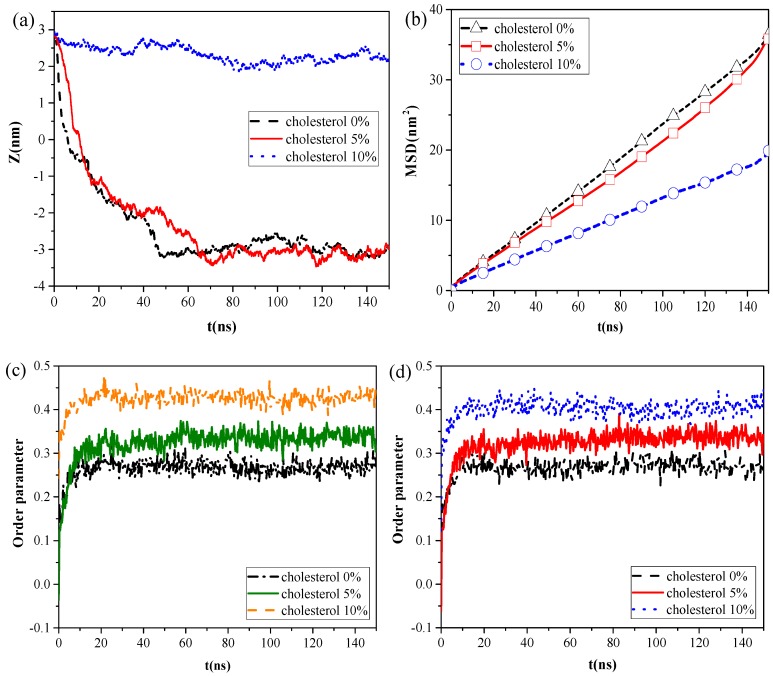
Effect of cholesterol content of the monolayer: (**a**) centroid–centroid distance between the CaSO_4_ NP and the monolayer, (**b**) MSDs of lipid molecules in the CaSO_4_ NP monolayer system, and (**c**,**d**) order parameters for the systems without NPs and with the CaSO_4_ NP.

**Figure 5 ijms-20-03281-f005:**
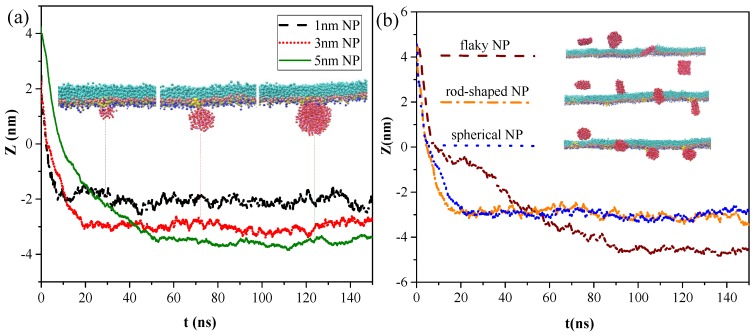
Snapshot of transmembrane and centroid-to-centroid distances of NPs with different (**a**) NP sizes and (**b**) NP shapes.

**Figure 6 ijms-20-03281-f006:**
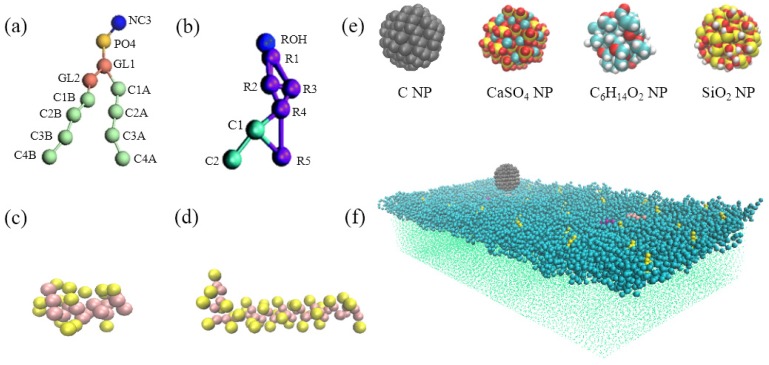
Coarse-grained (CG) models of (**a**) the dipalmitoylphosphatidylcholine (DPPC) molecule; (**b**) cholesterol molecule; (**c**) SP-B protein; (**d**) SP-C protein; (**e**) C, CaSO_4_, C_6_H_14_O_2_, and SiO_2_ NPs; and (**f**) NP (cyan)- pulmonary surfactant monolayer with DPPC (blue), cholesterol (yellow), SP-B (purple), SP-C (pink), and water molecules (green).

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
