# Peer review of "A Simulation Study on the Interaction Between Pollutant Nanoparticles and the Pulmonary Surfactant Monolayer"

_ijms, 2019, doi:10.3390/ijms20133281_

Reviewer 1 Report

The manuscript of Yue et al. reports the results from a coarse-grained MD study focusing on pollutant nanoparticles and how they interact with the pulmonary surfactant. Given the interesting and important topic addressed and the methods used, I would like to recommend publication of this manuscript in Int. J. Mol. Sci., once a few points have been addressed:

1. The authors should give some information on the rationale for choosing these four nanoparticles.

2. Since the readership of molecules is interdisciplinary, the authors should provide some background information about MD simulations and their applications in the introduction. I would like to recommend the following reviews: 10.1146/annurev-biophys-042910-155245 or 10.1016/j.drudis.2015.01.003. In this context it should also be clearly outlined why CG-MD was used and what are advantages and limitations of this method compared to classical MD.

3. The analyses contains several calculated values; the authors should discuss how coarse-graining might affect the outcome. Were the simulations repeated with random initial velocities (triplicates)?

4. In the methods part, it should be clearly stated which parameters were different for exhalation and inhalation states.

5. In figure 2b, CaSO4 shows a similar behavior like C6H14O2 and SiO2. I wonder why this similarity is not reflected in figure 2d, but in figure 2f.

Author Response

We really appreciate reviewers for their instructive comments, which are very valuable and helpful for improving our manuscript.

Comments and Suggestions for Authors

The manuscript of Yue et al. reports the results from a coarse-grained MD study focusing on pollutant nanoparticles and how they interact with the pulmonary surfactant. Given the interesting and important topic addressed and the methods used, I would like to recommend publication of this manuscript in Int. J. Mol. Sci., once a few points have been addressed:

1. The authors should give some information on the rationale for choosing these four nanoparticles.

AWe revised the sentence on page 10, line 336-340 as " Given that organic carbon, elemental carbon, SO42- and NO3- are the most abundant chemical components of PM2.5,30 we selected C NP, C6H14O2 NP and CaSO4 NP to represent the elemental carbon, alcoholic content, and secondary sulfate particulates discharged from automobile exhaust, coal combustion and biomass burning, and selected SiO2 NP to represent the main inorganic component of wind-blown soils. "

[30] Cao J J, Shen Z X, Chow J C, et al. Winter and Summer PM2.5 Chemical Compositions in Fourteen Chinese Cities[J]. Journal of the Air & Waste Management Association, 2012, 62(10):1214-1226.

2. Since the readership of molecules is interdisciplinary, the authors should provide some background information about MD simulations and their applications in the introduction. I would like to recommend the following reviews: 10.1146/annurev-biophys-042910-155245 or 10.1016/j.drudis.2015.01.003. In this context it should also be clearly outlined why CG-MD was used and what are advantages and limitations of this method compared to classical MD.

AThanks for the helpful suggestion. We added the sentence " Some simulations have been performed using molecular dynamics method, which is widely used to compute biomolecular interactions at micro/nanoscale by determining the positions and velocities of particles in terms of the laws of classical physics.14 " in the "Introduction" section, on Page 2, line 52-55.

We revised as " All simulations were carried out using the MARTINI27,28 CG force field, in which several heavy atoms were incorporated into one bead to decrease computational cost and allow larger length scale and longer time-period simulations compared with all-atom models. The coarse-grained procedure would reduce the resolution of the system and might cause some key information at the atomic scale to be lost.29 ", to state advantages and limitations of CG MD in the " Method " section, on Page 10, line 327-331.

[14] Dror R O, Dirks R M, Grossman J P, et al. Biomolecular Simulation: A Computational Microscope for Molecular Biology[J]. ANNUAL REVIEW OF BIOPHYSICS, VOL 41, 2012, 41(1):429-452.

[29]Saunders M G, Voth G A. Coarse-Graining Methods for Computational Biology[J]. Annual Review of Biophysics, 2013, 42(42):73-93.

3. The analyses contains several calculated values; the authors should discuss how coarse-graining might affect the outcome. Were the simulations repeated with random initial velocities (triplicates)?

AWe added " In our simulation, the model of pulmonary monolayer was established using the standard coarse-grained MARTINI force field and the coarse-graining of NPs was performed by using one bead to represent one atom for reducing the loss of information. " in the " Method " section, on Page 10, line 331-333. And yes, we repeated the simulations in triplicate under same conditions and found the difference is slight, which did not affect the results of analysis.

4. In the methods part, it should be clearly stated which parameters were different for exhalation and inhalation states.

AThanks for reviewer’s suggestion. We revised the sentence as " a large lateral pressure was applied in the simulation to achieve the surface tension of the pulmonary surfactant monolayer in the respiration process, which was 30% higher and lower than that at initial state for the exhalation (compression) and inhalation (expansion) breathing states, respectively." to state the parameters, which is different in exhalation and inhalation states, on page 11, line 365-368.

5. In figure 2b, CaSO4 shows a similar behavior like C6H14O2 and SiO2. I wonder why this similarity is not reflected in figure 2d, but in figure 2f.

A: " The MSD is used to evaluate the movement of a given particle, indicating its diffusion rate.23 " We added this description in the “Results and Discussion” section on page 2, line 80-81.

The MSD is affected by the hydrophilic/hydrophobic properties of NPs. Although the semi-hydrophilic CaSO4 NP moved into the monolayer much more slowly than the strongly hydrophilic C6H14O2 NP and SiO2 NP, as shown in Fig. 2(d) (Fig. 1(d) in the revised manuscript)), it can still penetrate the monolayer when the system reached equilibrium. So the centroid-to-centroid distance between the CaSO4 NP and monolayer in Fig. 2(f) (Fig. 1(f) in the revised manuscript) shows similar behavior like C6H14O2 and SiO2 as that in Fig.2(b) (Fig. 1(b) in the revised manuscript).

Reviewer 2 Report

This manuscript reports a simulation study of the interaction between different nanoparticles (NP) and a surfactant monolayer carried out with a coarse grained molecular model adopting the MARTINI force field.

The subject is certainly of interest, while the simulations appear to be properly carried out and the results appropriately analyzed and discussed, so that I believe that the manuscript could be accepted with minor changes only.  The main issues and/or observations are listed below in the order they apply to the text.

1. At lines 69-70 the authors claim that " the simulation results were compared with experimental data from literature to validate the simulation model", but only a very qualitative comparison is shown in Fig. 4e.  Therefore this statement should be definitely toned down.

2. The formula C6H14O2 may correspond to different compounds: most likely, the authors refer to one hexanediol isomer (but which one?), but these compounds are water-soluble. So how could it form a nanoparticle in real experiments?

3. In sect. 2, the (average) thickness of the monolayer should be clearly given, even though it is mentioned at line 156.  At the end of this section, the statement at lines 93-95 is unclear to me: if PBC were used and there is a vacuum above the monolayer and below the water slab in the z direction (Fig. 1f), how could the water molecules move from "below" to "above" the monolayer?

4. At p. 4, in Figs. 2c and 2d the units of MSD should be given. Moreover, the initial position of the NP at time t=0 should be given in terms of the distance from the monolayer surface: from the snapshots of Figs. 2a and 2b some displacement is evident, unlike what shown by Figs. 2c and 2d for the CaSO4 NP and the C6H14O2 NP at least up to 100 ns.

5. In Fig. 3 the order parameter is plotted in "arbitrary units": why, in view of eq. 2, which gives definite values in the 0 - 1 range?

6. In Fig. 4a what is meant by the histogram labeled as "before simulation"? And why does it become larger (i.e., less negative) after interaction between the NP and the monolayer? And what is the NP energy in Fig. 4b? According to what said at line 86, I gathered that the NP energy was basically a constant independent of compression or expansion.  Finally, in this context some remarks about the extent of hydrogen bonding of the hydrophilic nanoparticles with the polar heads of DPPC and/or with water would be useful.

7. In sect. 3.3, the authors mention simulation results obtained with a varying amount of cholesterol within the monolayer, but report only the results with the CaSO4 NP: why the results with the other NPs are not mentioned? And which is the hydrophilic NP discussed at p. 10 and shown in Fig. 6a and 6b? Incidentally, note that at line 350 the flaky NP is most likely a misprint for the rod-shaped NP.

Author Response

We really appreciate reviewers for their instructive comments, which are very valuable and helpful for improving our manuscript.

This manuscript reports a simulation study of the interaction between different nanoparticles (NP) and a surfactant monolayer carried out with a coarse-grained molecular model adopting the MARTINI force field.

The subject is certainly of interest, while the simulations appear to be properly carried out and the results appropriately analyzed and discussed, so that I believe that the manuscript could be accepted with minor changes only. The main issues and/or observations are listed below in the order they apply to the text.

1. At lines 69-70 the authors claim that " the simulation results were compared with experimental data from literature to validate the simulation model", but only a very qualitative comparison is shown in Fig. 4e. Therefore this statement should be definitely toned down.

A: Thanks for the helpful suggestion. We revised the sentence as: " and a qualitative comparison of the simulation results and the experimental data from literature was carried out " in the " Introduction " section, on Page 2, line 71-72.

2. The formula C6H14O2 may correspond to different compounds: most likely, the authors refer to one hexanediol isomer (but which one?), but these compounds are water-soluble. So how could it form a nanoparticle in real experiments?

A: As reviewer mentioned, C6H14O2 NP is water-soluble and may correspond to different compounds. One of them is pinacol, which exists in the form of solid at normal temperature. So, we think the NP might be possibly formed if there are not sufficient quantities of water to completely dissolve it. We were supposed to select a representative organic particle, and thanks for reviewer’s reminder, we will furtherly confirm it by experiment. 

3. In sect. 2, the (average) thickness of the monolayer should be clearly given, even though it is mentioned at line 156.  At the end of this section, the statement at lines 93-95 is unclear to me: if PBC were used and there is a vacuum above the monolayer and below the water slab in the z direction (Fig. 1f), how could the water molecules move from "below" to "above" the monolayer?

A: We gave the average thickness of the monolayer (2.08nm) on Page 3, line 114.

The velocity of a particle in molecular dynamics is randomly generated, so it can move in all directions. For PBC, in order to ensure that the number of particles in the simulation system is kept constant, the particles must return from the opposite interface to the model if they run out of the model during movement. So, the particles will re-enter the box from the " above " if they run out of the " below " of the box.

4. At p. 4, in Figs. 2c and 2d the units of MSD should be given. Moreover, the initial position of the NP at time t=0 should be given in terms of the distance from the monolayer surface: from the snapshots of Figs. 2a and 2b some displacement is evident, unlike what shown by Figs. 2c and 2d for the CaSO4 NP and the C6H14O2 NP at least up to 100 ns.

A: Thanks for the helpful suggestion. We added " The initial centroid-to-centroid distance of the NP and the monolayer at time t=0 was set to 3.2 nm" on Page 3, line 109-110.

We added the units of MSD in Figs. 2(c), 2(d) and Fig. 5(b) (Figs. 1(c), 1(d) and Fig. 4(b) in the revised manuscript) as “nm2. The MSDs in Figs. 2(c) and 2(d) (Figs. 1(c) and 1(d) in the revised manuscript) show the diffusion rate of the particles, rather than the displacement of the particles, as described in literature 23. We added a description of the MSD on page2, line 80-81. It can be seen from the figure that the diffusion rate of hydrophilic NPs is faster. The positional changes of the particles are given in Figs. 2(e) and 2(f) (Figs. 1(e) and 1(f) in the revised manuscript), and it can be seen that the displacement of the particles is evident.

5. In Fig. 3 the order parameter is plotted in "arbitrary units": why, in view of eq. 2, which gives definite values in the 0 - 1 range?

A: Sorry, we made a mistake about the unit of the order parameter. In fact, it is dimensionless, so we deleted the unit of the order parameter in Fig. 3(a) and 3(b) (Fig. 2(a) and 2(b) in the revised manuscript). According to eq.2, the range of the order parameter is (-, 1). represents the angle between the bond linking two CG beads and z-axis, and these two beads are DPPC C3B and C4B, as shown in Fig. 1(a) (Figs. 6(a) in the revised manuscript). Because the change in this angle is not extremely large, the range of the order parameter is (0, ) in our simulation.

6. In Fig. 4a what is meant by the histogram labeled as "before simulation"? And why does it become larger (i.e., less negative) after interaction between the NP and the monolayer? And what is the NP energy in Fig. 4b? According to what said at line 86, I gathered that the NP energy was basically a constant independent of compression or expansion.  Finally, in this context some remarks about the extent of hydrogen bonding of the hydrophilic nanoparticles with the polar heads of DPPC and/or with water would be useful.

A: We revised the "before simulation" as "initial state" in Fig. 4(a) (Fig. 3(a) in revised manuscript). Because the change in total energy is equal to the change in potential energy, as described on page 6, line 188, and the potential energy is converted to the kinetic energy of the particle moving toward the monolayer, as described page 6, line 193, so the potential energy of the system is reduced, that is, the total energy is decreased. We added the sentence " so the potential energy of the system was decreased. " on Page 6, line 195.

We revised the expression of the Y-axis. "Fig. 3(b) shows the difference in total energy between the NP-monolayer system and the pure monolayer system, which includes the energy of the NP, the interaction energy between the NP and monolayer, and the interaction energy between the NP and water molecules." as shown on Page 6, line 197-199. And we revised Fig. 3(b) as "ΔEtotal (i.e. the difference in total energy between the NP-monolayer system and the pure monolayer system) " on page 7, line 241-243.

As reviewer mentioned, the description of the hydrogen bond is meaningful, but in the coarse-grained force field, the hydrogen bonding is considered as a non-bond interaction, which is included in the potential energy of the system and cannot be described separately.

7. In sect. 3.3, the authors mention simulation results obtained with a varying amount of cholesterol within the monolayer, but report only the results with the CaSO4 NP: why the results with the other NPs are not mentioned? And which is the hydrophilic NP discussed at p. 10 and shown in Fig. 6a and 6b? Incidentally, note that at line 350 the flaky NP is most likely a misprint for the rod-shaped NP.

A: The results show that C NP cannot cross the monolayer at all three levels of cholesterol contents whereas C6H14O2 NP and SiO2 NP could penetrate the monolayer.

So the effect of cholesterol content cannot be clearly analyzed using these three NPs. The reason for selecting CaSO4 NP is that cholesterol content has an obvious effect on penetration of CaSO4 NP.

Because hydrophilic NPs can cross the monolayer, so we chose hydrophilic NPs to analyze the effects of NP shape on NP transmembrane. We revised the "flake NP" in Fig. 6(b) (Fig. 5(b) in revised manuscript) as " flaky NP " and revised " flaky NP " as " rod-shaped NP " on Page 9, line 320.